# Analysis of Microbial Communities in Aged Refuse Based on 16S Sequencing

**Fen Hou [1],\*, Junjie Du [2], Ye Yuan [1], Xihui Wu [1] and Sai Zhao [1]**

1   Department of Land Resource Management, School of Public Administration,
    Shanxi University of Finance and Economics, Taiyuan 030012, China; yuanye2452@cugb.edu.cn (Y.Y.);
    20161058@sxufe.edu.cn (X.W.); zhaosai@yinzhaowang.com (S.Z.)
2   College of Life Science, Shanxi Normal University, Linfen 710062, China; junjiedu@sxnu.edu.cn
\*   Correspondence: 20181064@sxufe.edu.cn

**Abstract:** Aged refuse is widely considered to have certain soil fertility. 16S rRNA amplicon sequencing is used to investigate the microbial community of aged refuse. The aged refuse is found to contain higher soil fertility elements (total nitrogen, total phosphorus, total potassium, etc.) and higher concentrations of heavy metals (Pb, Cd, Zn, and Hg). Taxonomy based on operational taxonomic units (OTUs) shows that *Actinobacteria*, *Proteobacteria*, *Chloroflexi*, *Acidobacteria*, and *Gemmatimonadetes* are the main bacterial phyla in the two soils and there is a palpable difference in the microbial community composition between the two groups of samples. The genera *Paramaledivibacter*, *Limnochorda*, *Marinobacter*, *Pseudaminobacter*, *Kocuria*, *Bdellovibrio*, *Halomonas*, *Gillisia*, and *Membranicola* are enriched in the aged refuse. Functional predictive analysis shows that both the control soil and aged refuse have a high abundance of "carbohydrate metabolism" and "amino acid metabolism", and show differences in the abundance of several metabolism pathways, such as "xenobiotics biodegradation and metabolism" and "lipid metabolism". Aged refuse and undisturbed soil show significant differences in alpha diversity and microbial community composition. Multiple environmental factors (Hg, TN, Cr, Cd, etc.) significantly impact microorganisms' abundance (*Marinobacter*, *Halomonas*, *Blastococcus*, etc.). Our study provides valuable knowledge for the ecological restoration of closed landfills.

**Keywords:** 16S rRNA amplicon; aged refuse; fertility elements; heavy metals; microbial community; microbial diversity

## 1. Introduction

As the most populous country at present, China is facing tremendous pressure with regard to appropriate methods of municipal solid waste (MSW) disposal. In 2019, China's population reached 1.4 billion, and the amount of MSW removed and transported reached 242 million tons (China's National Bureau of Statistics, 2020). Nearly a quarter of MSW is disposed of at landfills [1]. The reduction of annual MSW production and the ecological treatment of historically accumulated waste landfills have become increasingly urgent [2,3].

Previous studies have shown that after approximately 10 years of burial, the degradable substances in a landfill are fully broken down, and the natural gas or leachate has reached a low level [4,5]. The aged refuse at this stabilized or mature stage has the potential to be utilized in many ways; for instance, it has certain characteristics of fertile soil and can be used for the adsorption treatment of pollutants [6,7]. In recent years, there have been many reports on the use of aged refuse in landfill leachate, water pollution, and biological reaction vessels [6,8–10]. For instance, reports have shown that it can be used to promote anaerobic catabolism in kitchen waste, as well as the harmless treatment of soil contaminated by total petroleum hydrocarbons [9,10]. Different methods have also been developed for waste treatments or reuse. For example, anaerobic membrane bioreactors and mixed-matrix membranes had been reported to have high potential in food industrial wastewater and municipal wastewater treatment [11,12].

As an important biologically active component of soil, the microbial community of aged refuse should also be explored [13]. A previous study showed that there were some differences between the microbial composition of in-use and closed landfills [14]. Genes related to nitrogen metabolism by microorganisms play an important role in the use of mineralized waste to treat landfill leachate [15]. There are also reports on the microbial community and functional composition of landfills in New Delhi, India [16,17]. However, in general, these studies on landfill microbiology have many limitations, and more in-depth research is urgently needed [18]. Soil microorganisms are important components of natural and managed ecosystems, containing diverse individual microbial taxa, which play crucial roles in the maintenance of soil fertility, nutrient cycling, and soil carbon sequestration [19]. Studies have reported the effect of soil microorganisms and their metabolic function for soil fertility, demonstrating how the management of soil microorganisms contributes to improved soil quality [19,20]. Soil bacteria (*Rhizobium*) form a symbiotic relationship with roots and participate in biological nitrogen fixation [21]. For example, *Sinorhizobium* sp. and *Rhizobium* sp. are two soil bacteria that can nodulate with alfalfa plants [22].

In this study, aged refuse from the surface soil of a stable landfill and natural soil from nearby areas without human intervention are excavated for high-throughput sequencing of 16S rRNA amplicons. This study aims to explore the differences in microbial composition between mineralized soil and normal soil and to further explore the relationship between microbes and soil fertility, heavy metal pollution, electrical conductivity, and other indicators.

## 2. Materials and Methods

### 2.1. Soil Sample Acquisition

After sealing a landfill for 8–10 years, the surface settlement is very small, the natural leachate and gas volume of the landfill is little or none, and the easily degradable substances in the landfill are completely or nearly completely degraded and decomposed into more stable soil-like substances, that is, mineralized refuse. Three samples of mineralized refuse (also named aged refuse) soils were excavated at a depth of 15 cm from the Xingou municipal solid waste landfill, which is located in the east of Taiyuan in China with a maximum capacity of 3.5 million cubic meters, opened in 1987 and closed in 2007. The landfill is a gully type on the Loess Plateau. Three samples of natural surface soil from nearby pollution-free, non-fertilized natural topsoil were excavated and used as controls. The control samples were collected from the area within 5 km of the landfill. According to the international soil classification World Reference Base (WRB), the control soil belongs to calcareous highly active leached soil, and the soil horizons are calcic horizons.

Samples were collected according to the principle of "random, multi-point, uniform" sampling, as described in the study of Bao et al. [23]. A total of 15 points were sampled in a snake shape, and the samples from five points were mixed as one sample. Before sampling, the surface sundries were removed and a 15 cm deep pit was dug. Then, a soil slice of about 5 cm thick was evenly cut on the vertical pit wall with a soil shovel. The sampling methods and depth of soil samples taken at each sampling point were consistent between the aged refuse and control soil. For both the control soil and aged refuse samples, the non-degradable impurities of a large size, such as stones, glass bottles, plastic films, and rubbers, were removed after air-drying, ground granulation, and screening through a 20-mesh screen (900 pm). The fine material was sieved into five different particle size ranges of 900–300, 300–150, 150–105, 105–90, and 90–0 μm. The largest particle size (900–300 μm) group had the largest mass ratio at 39.6%; the smallest particle size (90–0 μm) group had a mass ratio of 28.4%; the remaining groups had ranges of 300–150, 150–105, and 105–90 μm, accounting for 20.7%, 5.8%, and 5.4% of the mass, respectively.

### 2.2. Physical and Chemical Indicator Measurements

The mercury (Hg), arsenic (As), and lead (Pb) contents were determined using atomic fluorescence spectrometry performed on an atomic fluorescence spectrophotometer. The

chromium (Cr), nickel (Ni), cadmium (Cd), and zinc (Zn) contents were determined using aqua regia extraction–inductively coupled plasma mass spectrometry performed on an inductively coupled plasma source mass spectrometer. The pH was measured using the pH glass electrode method performed on a pH meter. Cation exchange capacity was measured using ammonium acetate exchange–inductively coupled plasma emission spectrometry performed on an inductively coupled plasma source mass spectrometer. Conductivity was measured using conductometry performed on a portable digital multi-parameter analyzer. Soil organic carbon (SOC) was measured using potassium dichromate-sulfuric acid oxidation. Total potassium (TK) was measured using atomic absorption spectrophotometry performed on an atomic absorption spectrophotometer. Total phosphorus (TP) was measured using inductively coupled plasma emission spectroscopy performed on an inductively coupled plasma emission spectrometer. Total nitrogen (TN) was measured by the Kjeldahl, ascertained using automatic Kjeldahl apparatus.

### 2.3. DNA Extraction and High-Throughput Sequencing

The soil DNA was extracted and prepared using an E.Z.N.A.® Soil DNA Kit (Omega Bio-Tek Inc. Guangzhou, China) according to the manufacturer's instructions. DNA was dissolved in a Tris- ethylene diamine tetraacetic acid (EDTA) buffer and stored in a $-80\,^{\circ}$C ultra-low temperature refrigerator. The concentration of the DNA was determined using Agilent Bioanalyzer 2100. NEXTflex® 16S Amplicon-Seq Kits were used for library preparation. Sequencing was performed using the Illumina MiSeq® sequencing platform. Each sample generated at least 15,000 paired-end reads.

### 2.4. High-Throughput Sequencing Data Analysis

The analysis of 16S rRNA amplicon data for the soils was mainly performed using QIIME toolkits [24]. Based on the UCLUST algorithm, a 97% similarity level clustering was performed to generate an operational taxonomic units (OTU) sequence set [25]. The Greengenes database (Release 13.8, http://greengenes.secondgenome.com/, accessed on 19 October 2020) was used for taxonomic classification and annotation. Chao1, Shannon, Simpson, and abundance-based coverage estimators (ACE) indices were used to evaluate the diversity level of the samples. Principal component analysis (PCA) analysis, Pearson's correlation coefficient, and permutational multivariate analysis of variance (PERMANOVA) were performed based on the "vegan" R package. The Metastats method, based on Fisher's exact test and the non-parametric t-test, was used to analyze the taxonomy in terms of the significant differences between the groups [26]. PICRUSt was used for functional predictive analysis [27].

### 2.5. Statistical Analysis

All quantitative data conformed to the normal distribution using the Shapiro–Wilk test. Differences between groups were compared using an independent sample t-test. Statistical analysis was performed using SPSS software (version 19, IBM, Armonk, NY, USA).

## 3. Results

### 3.1. Comparison of Physical and Chemical Factors

The changes in all physical and chemical indicators of the aged refuse group and the control group are shown in Figure 1. Foremost, in terms of the three main total nutrient elements, it can be seen that the TN and phosphorus concentrations of the aged refuse group were significantly higher than those of the control group ($p < 0.001$). In contrast, the total potassium concentration in the control group was significantly higher ($p < 0.001$) (Figure 1A). A similar significant increase in the soil organic carbon content of the aged refuse group was detected ($p < 0.05$) (Figure 1B).

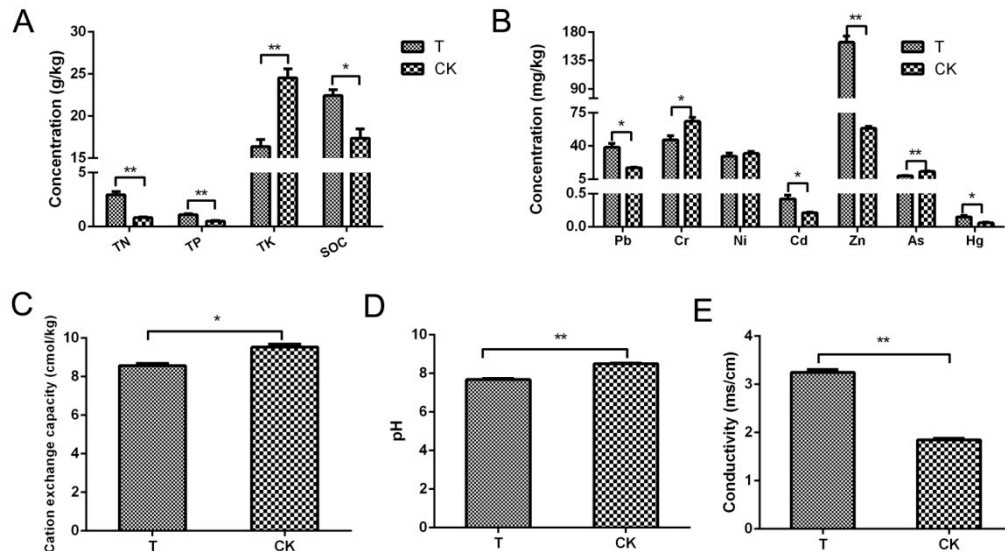

**Figure 1.** Physical and chemical indexes of aged refuse and control soil. (**A**) Concentrations of the total nitrogen (TN), total phosphorus (TP), total potassium (TK), and soil organic carbon (SOC) in the two groups. (**B**) Concentrations of primary metallic or harmful elements in the two groups of soils. (**C**) Cation exchange capacities, (**D**) pH values, and (**E**) conductivity values of the two groups. CK and T represent the control and aged refuse groups, respectively. ** $p < 0.001$, * $p < 0.05$ (t-test or Mann–Whitney test).

There were also many statistically significant differences in the concentrations of heavy metals or harmful elements between the two groups of soil. Excluding nickel, the remaining six heavy metal elements that were detected showed significant differences. The aged refuse group contained higher levels of lead, cadmium, zinc, and mercury with a significant level of difference ($p < 0.05$). In contrast, the control group had significantly higher levels of chromium and arsenic elements ($p < 0.05$) (Figure 1C). Additionally, there were statistically significant differences between the two groups of samples in terms of pH and physical indicators ($p < 0.05$) (Figure 1D,E).

### 3.2. Microbial Community Structure Composition

16S amplicon sequencing based on the Illumina MiSeq platform produced a total of 287,749 high-quality paired-end reads in a total of six samples. A total of 9936 high-quality OTUs were obtained from a total of 59,495 raw OTUs, which had been clustered from merged paired-end reads. Compared with the control group, the global microbial composition of the three samples in the aged refuse group was relatively consistent (Figure 2A). Overall, the main taxonomic abundance comprised the bacteria of the five phyla (*Actinobacteria*, *Proteobacteria*, *Chloroflexi*, *Acidobacteria*, and *Gemmatimonadetes*), with a relative abundance of more than 80%, in both of the groups of samples (Figure 2B). The control group had a higher relative abundance of *Actinobacteria* and *Acidobacteria*, while the aged refuse group had a higher proportion of *Proteobacteria*, *Bacteroidetes*, *Patescibacteria*, and *Firmicutes* (Figure 2B). The relative abundances at the genus level also showed visible differences in the species composition of the two groups (Figure 2C). More detailed taxa with statistically significant differences at the genus level are shown in Figure 3. Abundances of genera such as *Paramaledivibacter*, *Limnochorda*, *Marinobacter*, *Pseudaminobacter*, *Kocuria*, *Bdellovibrio*, *Halomonas*, *Gillisia*, and *Membranicola* were higher in the aged refuse group. The genera of *Lechevalieria*, *Solirubrobacter*, *Blastococcus*, *Reyranella*, *Neo-b11*, *Geodermatophilus*, *Ellin6055*, *Nitrospira*, *CL500-29_marine_group*, *Ellin6067*, and *Gaiella* were enriched in the control group.

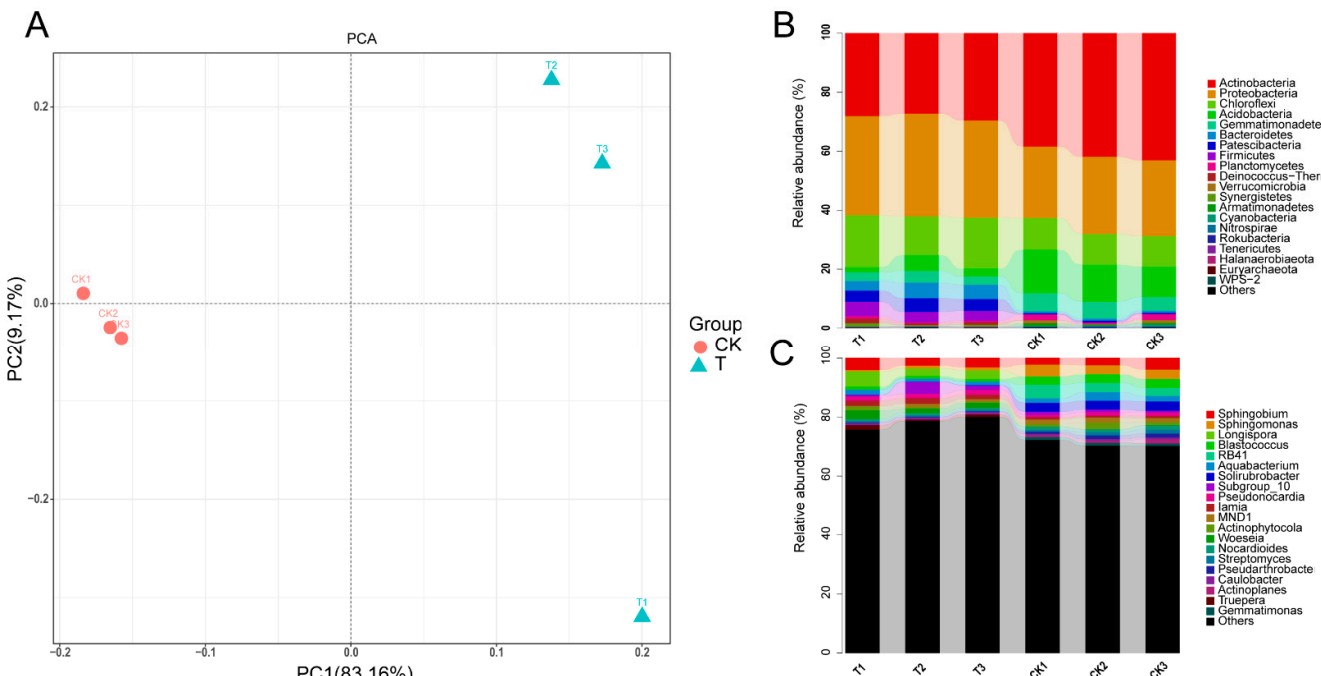

**Figure 2.** The global microbial composition. (**A**) PCA analysis based on relative abundance at the genus level shows the similarity of community structure in the control and aged refuse groups. The first and second axes explained 83.16% and 29.17% of the variance, respectively. The top 20 predominant microbial phyla (**B**) and top 20 predominant microbial genera (**C**) of the two groups are shown. The relative abundance of each microbial phyla or microbial genera were different between the control and aged refuse group. CK and T represent the control and aged refuse groups, respectively.

### 3.3. Diversity of the Microbial Community

On the species level, the within-sample (alpha) diversity was calculated on the basis of Chao1, Shannon, Simpson, and ACE indices to evaluate the microbial richness and evenness of both groups. As shown in Table 1, the Chao1 and ACE indices were higher in the aged refuse group, while the Simpson index was lower in the aged refuse group in comparison to that of the control group, indicating higher diversity of the microbial community in the aged refuse.

**Table 1.** The alpha diversity indices in the two groups.

| Samples | Simpson | Chao1 | ACE | Shannon |
|---|---|---|---|---|
| T1 | 0.993826 | 3118.17 | 3207.86 | 9.33 |
| T2 | 0.996125 | 3321.49 | 3708.06 | 9.69 |
| T3 | 0.995019 | 3411.14 | 3652.43 | 9.59 |
| CK1 | 0.998292 | 3049 | 3049 | 10.73 |
| CK2 | 0.997839 | 3000 | 3000 | 10.56 |
| CK3 | 0.997648 | 3349.22 | 3359.57 | 10.7 |

CK and T represent the control and aged refuse groups, respectively.

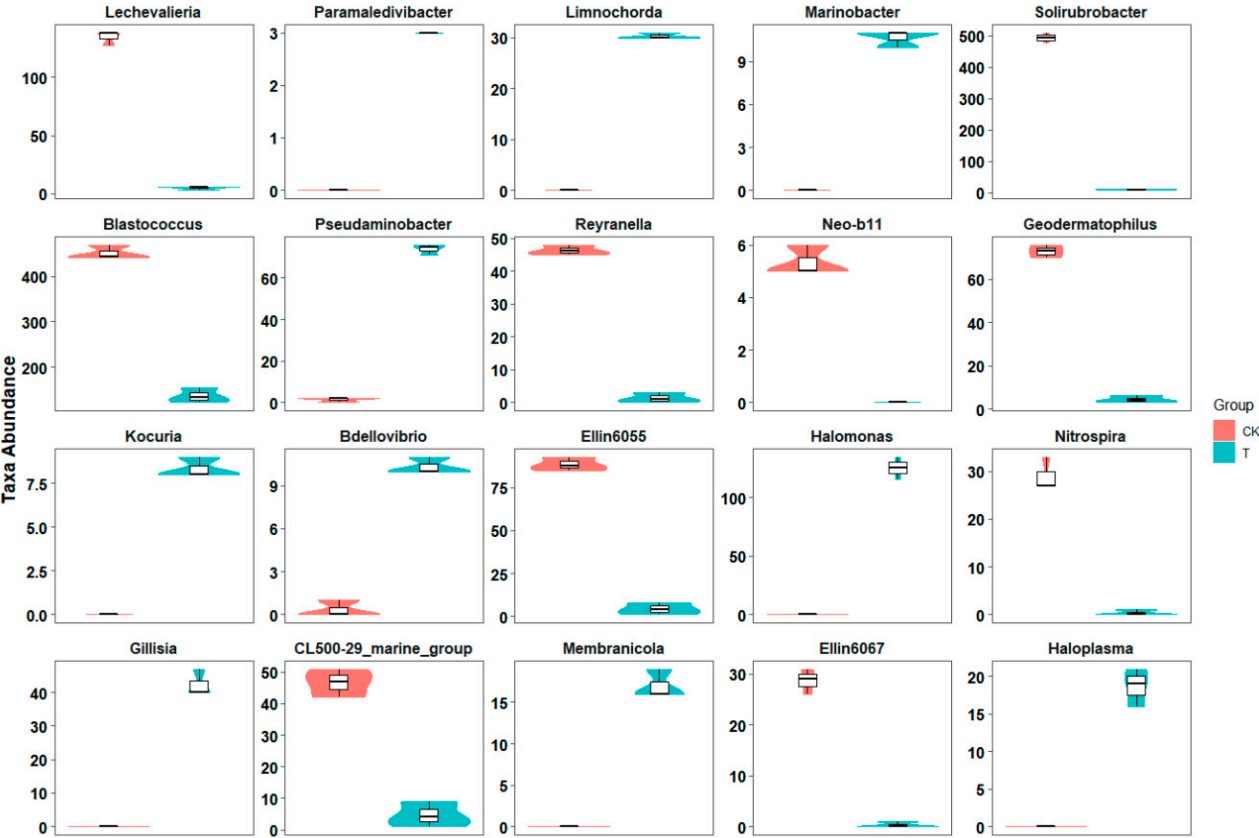

**Figure 3.** Difference in taxa abundance at the genera level. Boxplots show the taxa abundance at the genera level was significantly different between the control and aged refuse groups. The ordinate represents the abundance of the opposing genus in each group. The 20 genera with the most significant q-value are shown (Metastats, q-value < 0.05). CK and T represent the control and aged refuse groups, respectively.

Rarefaction plots of the alpha diversity of both the aged refuse and control groups were analyzed. When the sequence number of each sample reached 16,000, the two alpha-diversity indexes both reached a plateau (Figure 4). The Shannon index of the control group was significantly higher than that of the aged refuse group (Figure 4A; $p < 0.01$, t-test), although there was no significant difference in the Chao1 index of the two soil samples (Figure 4B). This was also consistent with the dilution curve results for the observed species (Figure 4C). Although the observed species index of the control group was significantly higher than that of the aged refuse, this difference became insignificant in the Chao1 index due to Chao1's correction algorithm for the number of low-abundance species. Regardless of the aged refuse or control group, the weight UniFrac distance between the groups was much greater than the distance within them (Figure 4D).

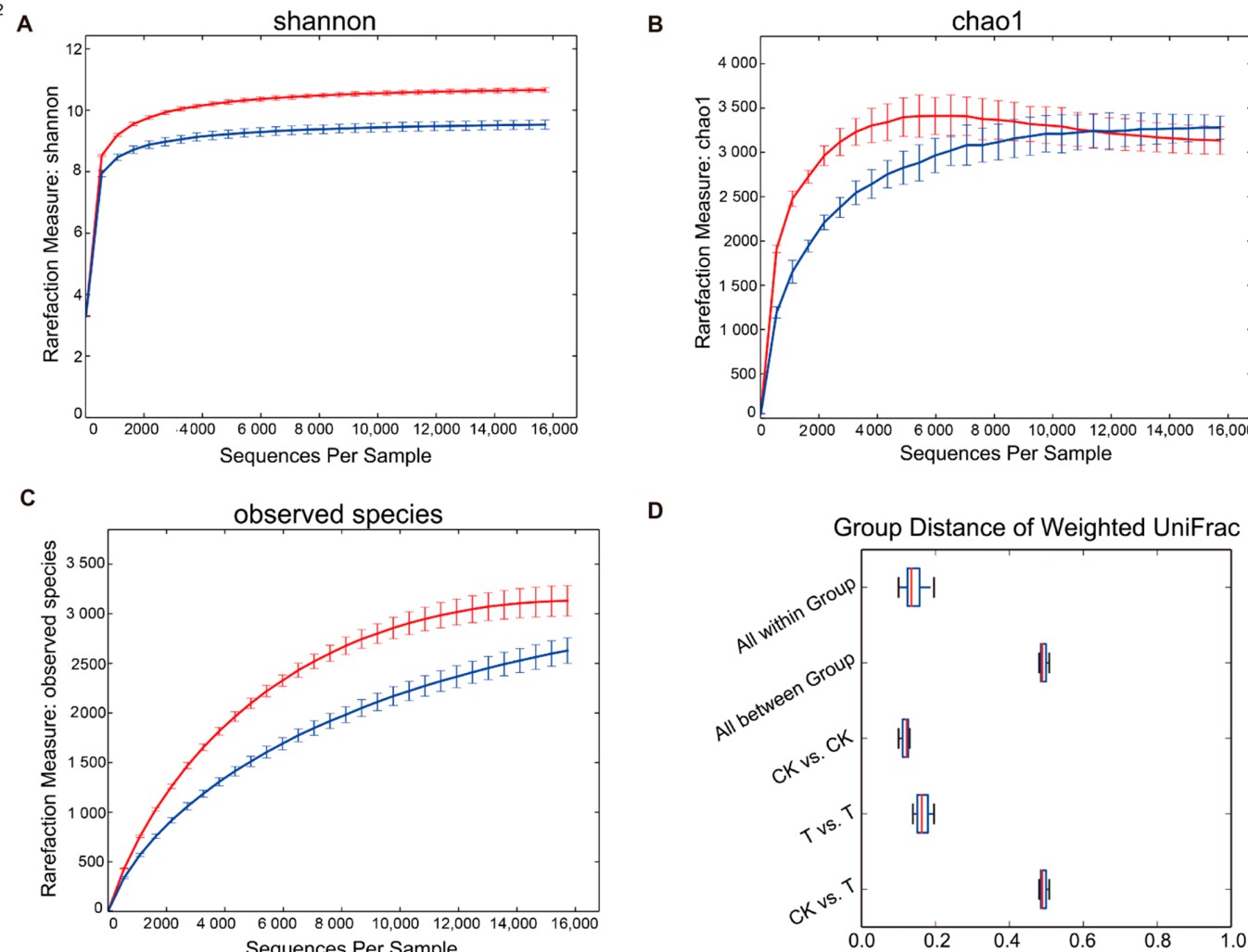

**Figure 4.** Rarefaction plots of two soil groups. The rarefaction plots based on (**A**) Shannon, (**B**) Chao1, and (**C**) observed species index show the alpha-diversity of the microbial community in the control and aged refuse groups. The red line represents the control group and the blue line represents the aged refuse group. Each point and error bar represents the mean and standard error of the alpha-diversity indexes, respectively. (**D**) The Weighted UniFrac distances between and within groups are displayed using boxplots. CK and T represent the control and aged refuse groups, respectively.

### 3.4. The Interactions between the Microorganisms and Physical or Chemical Indicators

Pearson's correlation analysis of genus-level abundance and environmental indicators showed that microorganisms, at least those that were significantly enriched in the two groups, had a strong correlation with these indicators (Figure 5). Further PERMANOVA results showed that the impact of multiple environmental factors on these microorganisms reached a statistically significant level of difference (Figure 5). For instance, it could be seen that *Marinobacter* was positively correlated with OP, Hg, AK, and TN, yet negatively correlated with Cr. Moreover, Pb and Cd were positively correlated with *Limnochorda* and *Halomonas*, yet negatively correlated with *Ellin6055* and *Blastococcus* ($p < 0.05$).

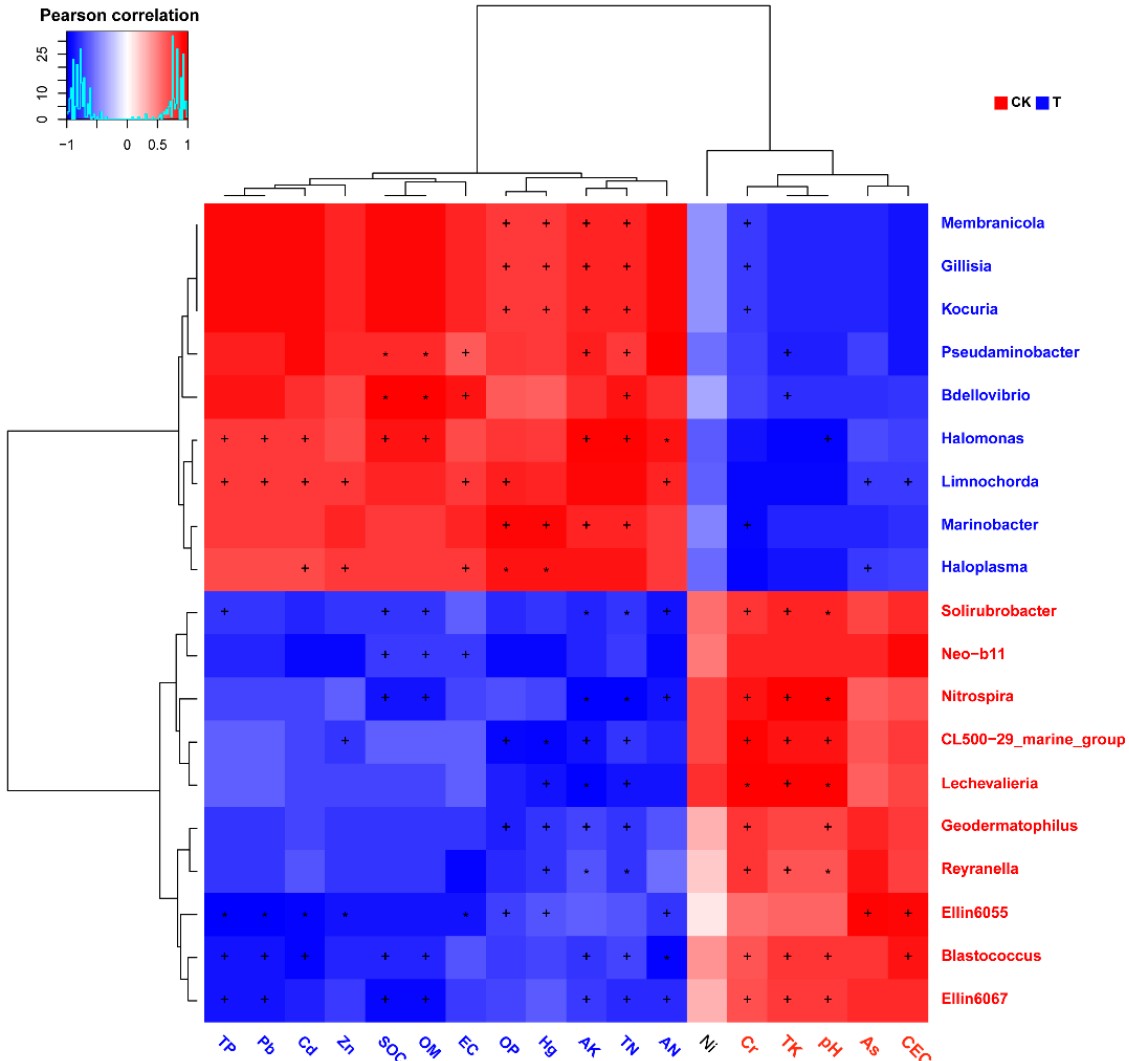

**Figure 5.** Heatmap of interactions between microorganisms and environmental factors. The X-axis represents the physical or chemical factors detected in this experiment. The Y-axis represents the top 20 genera with the most significant differences between the aged refuse and the control group. The colors of the axis labels represent the groups in which the measured value of the factor or the abundance of the genus were significantly increased (red for control group and blue for aged refuse). Each color block represents the Pearson correlation coefficient between the physical and chemical factors and microbial abundance. The red color block represents a positive correlation, while the blue color represents a negative correlation. Darker colors indicate stronger correlation coefficients. * $p < 0.01$, + $p < 0.05$ (PERMANOVA).

*3.5. The Predictive Function Analysis*

The relative abundance of the pathways under the metabolism class of the two groups is shown in Figure 6. It can be seen that both the control soil and aged refuse had a high abundance of "carbohydrate metabolism" and "amino acid metabolism", indicating that these two might be the predominant modes of metabolism activity for the microbes in soil. Differences in the abundance of "xenobiotic biodegradation and metabolism", "lipid metabolism", and "enzyme families" were observed between the two groups. This confirmed a difference in the abundance of the dominant bacterial community of the two groups.

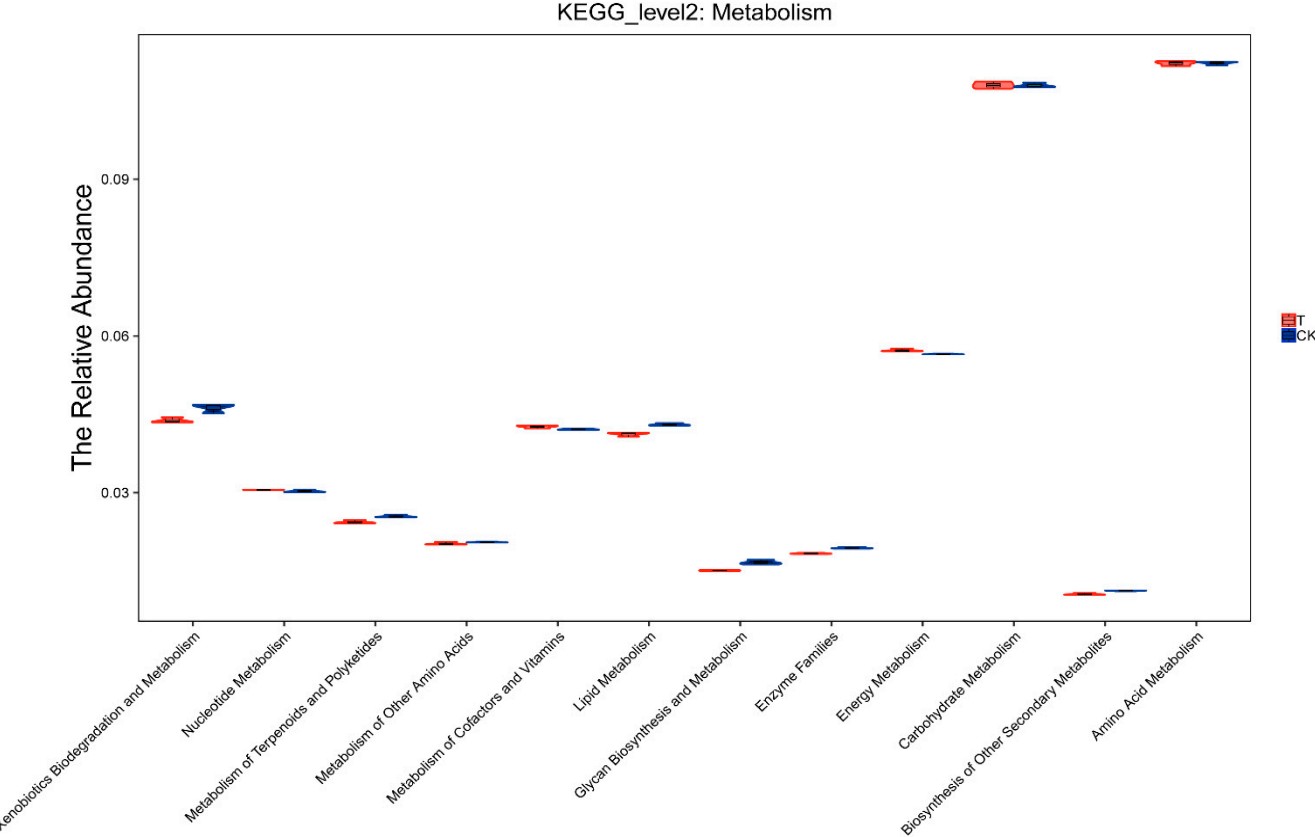

**Figure 6.** Relative abundance of metabolism. The violin plot shows the distribution of predicted KEGG_level 2 metabolism in the control and aged refuse groups. The X-axis represents the pathways under the metabolism class. The Y-axis represents the abundance of these pathways. CK and T represent the control and aged refuse groups, respectively.

## 4. Discussion

Owing to the difference in the background composition of the soil and the different types of pollutants, the chemical composition of each landfill varied greatly [28,29]. As age increases, the contents of carbon, nitrogen, phosphorus, and heavy metals in landfills gradually decrease until dynamic stability is achieved [5,29]. The concentrations of various metal elements in the aged refuse group, compared with the "Control standards of pollutants in sludge for agricultural use (CSPSA, GB 4284-2018)" and the "Soil environmental quality risk control standard for soil contamination of agricultural land (SCAL, GB 15618-2018)" are shown in Table A1. Heavy metal elements other than nickel, arsenic, and cadmium were more enriched in the mineralized waste. Generally, some heavy metal elements were enriched, although the composition of toxic elements in each landfill had its own unique features. The introduction of a landfill partially increased the concentration of harmful elements in the soil. However, over time, the content of these harmful elements dipped lower than the upper limit of the national requirements for agricultural land (Table A1).

The total potassium in the control soil was higher (Figure 1A); the potassium in the soil mainly came from the mineral potassium in insoluble minerals, which had a lower content in the original landfill waste. Many years after landfill, the contents of TN, phosphorus, potassium, and OM in the aged refuse were still higher than those of the control group (Figure 1B,C), which may be because of the biological or petrochemical OM contained in the original landfilled waste, such as kitchen waste, wood products, paper products, and bones. These results are consistent with those of previous studies [5,30,31]. As expected, the pH value of the aged refuse was lower than that of the control group (Figure 1E); that may be due to a relatively high content of OM, which promotes microbial acidification.



The cation exchange capacity of the control group was higher (Figure 1C) but was still at a relatively low level in both groups [32]. Most likely due to factors such as water content, OM content, and salt ions, the conductivity of the aged refuse group was higher (Figure 1E), which is considered to have a certain correlation with crop growth [33].

Alpha-diversity analysis showed that the two soil samples had similar Chao1 index values, but the Shannon index of the control group was higher (Figure 2A). This implies that the mineralized waste group contained more low-abundance bacterial species. Beta diversity analysis showed that although a great difference was seen between the two groups, the distance within the groups was smaller.

Species classification analysis revealed that there were four main bacterial phyla (the total abundance in any group exceeded 10%). Among these phyla, *Proteobacteria* and *Chloroflexi* were more abundant in the mineralized waste group (34% and 16% in the aged refuse group and 25% and 11% in the control group, respectively), and *Actinobacteria* and *Acidobacteria* had a higher abundance in the control group (28% and 3% in the aged refuse group and 41% and 12% in the control group, respectively) (Figure 2B). These differences show the huge impact of landfills on the soil microbial community's structure. These bacterial phyla are common in the environment and widely exist in various rivers, soils, oceans, and other environments. The microbial community in soil, one of the most complex microbial environments, varies greatly with the sampling site. There have also been great discrepancies in previous reports on microorganisms in landfill samples [14,16,17,34,35]. The microbiological analysis of landfills still needs to be discussed case-by -case. Therefore, more in-depth research on shotgun-based metagenomics or macrotranscriptomes of mineralized waste is urgently needed.

The correlation analysis between physical or chemical factors (Figure 5) and microbial abundance suggests that these specific microbial genera may have some special functions, such as heavy metal removal or carbon, nitrogen, and phosphorus metabolism. The relevance of some bacterial genera and pollutants (inorganic or organic) has been reported. Multiple studies have shown that *Marinobacter* species may be closely related to heavy metal resistance, such as to copper and lead [36–39]. *Solirubrobacter* and *Blastococcus* have been reported to be related to heavy metal tolerance [40], while *Membranicola* may promote the metabolism and tolerance of organic pollutants [8]. The results of these association analyses provide valuable information for subsequent verification. Soil microorganisms have been reported to alleviate soil polluted ecosystems. For example, Viesser et al. identified three rhizospheric bacteria—*Bacillus thurigiensis*, *Bacillus pumilus*, and *Rhodococcus hoagie*—in soil contaminated with petroleum, which could use petroleum hydrocarbons, as the sole carbon source and showed promising potential for petroleum biodegradation. In particular, *Rhodococcus hoagii* was highly efficient in consuming petroleum, reaching 87% within 24 h during degradation assays [41]. This study suggested that soil microorganisms have promising potential for alleviating soil-polluted ecosystems. The genus *Marinobacter* belongs to the family *Alteromonadaceae*, in *Proteobacteria*, which are Gram staining negative, flagellated, aerobic, and halophilic. Different bacteria that belong to the *Marinobacter* genus were isolated from oil-contaminated saline soil or saline soil and showed lipolytic activity [42,43]. The *Marinobacter* genus uses hydrocarbons and lipid compounds as specific substrates [44]. This genus was abundant in the aged refuse in this study, which seems to be the result of a broader physiological trait permitting the *Marinobacter* genus to consume lipids. In addition, metabolic function prediction analysis showed that the abundance of several metabolism pathways, such as "xenobiotics biodegradation and metabolism" and "lipid metabolism", were different between the two groups. As of this work, few research articles on the microbial diversity of solid aged refuse have been reported [13,14,16,18]. In particular, there has been no research on the difference between aged refuse and nearby undisturbed soils. In this study, the surface loam soils of a relatively stable landfill and nearby undisturbed natural soils were selected as the research objects. Multiple soil evaluation indicators have been measured, such as the concentrations of some toxic elements and OM. For further research, more time series and technically repeated

soils should be included, and the mining depth of the landfill sites should be more diverse. The interaction between organic pollutants and microorganisms requires more attention.

## 5. Conclusions

In this study, we found that the aged refuse contained higher soil fertility elements and higher concentrations of heavy metals. We further characterized the microbial composition and diversity of aged refuse and pollution-free and unfertilized soils using the 16S rRNA amplicon method. The aged refuse harbored a higher diversity of its microbial community, and the genera *Paramaledivibacter*, *Limnochorda*, *Marinobacter*, *Pseudaminobacter*, *Kocuria*, *Bdellovibrio*, *Halomonas*, *Gillisia*, and *Membranicola* were abundant in the aged refuse. Multiple environmental factors had significant impacts on microorganisms' abundance, including Hg, TN, Cr, and Cd. Pb and Cd were positively correlated with the abundance of *Limnochorda* and *Halomonas* yet negatively correlated with *Ellin6055* and *Blastococcus*. From both the microbial diversity index and the specific microbial community composition, it can be seen that aged refuse soils have their own unique patterns. Our study provides valuable knowledge for subsequent and more in-depth research on the treatment and restoration of closed landfills. The results indicate the feasibility of bioremediation in contaminated soil. In addition, they suggest that mineralized waste may be used as grassland soil to avoid land wastage caused by landfills and damage to the ecological environment. However, there is a need to select tolerant grass and to control exposure conditions. This will be investigated in the future.

**Author Contributions:** Data curation, F.H.; formal analysis, J.D.; investigation, F.H., Y.Y., and X.W.; methodology, F.H. and J.D.; resources, F.H. and S.Z. All authors have read and agreed to the published version of the manuscript.

**Funding:** This research project was supported by the doctoral research fund of Shanxi University of Finance and Economics.

**Institutional Review Board Statement:** Not applicable.

**Informed Consent Statement:** Not applicable.

**Data Availability Statement:** The data used in this study are available from the authors.

**Conflicts of Interest:** The authors declare no conflict of interest.

**Sample Availability:** The samples used in this study are available from the authors.

## Abbreviations

The following abbreviations are used in this manuscript:

| | |
|---|---|
| MSW | municipal solid waste |
| SOC | soil organic carbon |
| TK | total potassium |
| TP | total phosphorus |
| TN | total nitrogen |
| Hg | mercury |
| As | arsenic |
| Pb | lead |
| Cr | chromium |
| Ni | nickel |
| Cd | cadmium |
| Zn | zinc |
| CSPSA | control standards of pollutants in sludge for agricultural use |
| SCAL | soil contamination risk of agricultural land |

## Appendix A

**Table A1.** Concentrations of heavy metals (mg/kg) in the two soils and in the Chinese national standards.

| Metal (mg/kg) | Aged Refuse | Control | CSPSA Class A Upper Limit | CSPSAU Class B Upper Limit | SCAL Upper Limit |
|---|---|---|---|---|---|
| Pb | 38.48 ± 4.1 | 16.92 ± 0.92 | 300 | 1000 | 170 |
| Cr | 46.36 ± 4.53 | 65.89 ± 4.32 | 500 | 1000 | 250 |
| Ni | 29.12 ± 3.28 | 32.19 ± 2.01 | 100 | 200 | 190 |
| Cd | 0.42 ± 0.06 | 0.21 ± 0.01 | 3 | 15 | 0.6 |
| Zn | 163.83 ± 9.67 | 58.67 ± 1.81 | 1200 | 3000 | 300 |
| As | 8.45 ± 0.33 | 13.18 ± 0.38 | 30 | 75 | 25 |
| Hg | 0.15 ± 0.02 | 0.06 ± 0.01 | 3 | 15 | 3.4 |

CSPSA, control standards of pollutants in sludge for agricultural use GB 4284-2018; SCAL, soil contamination risk of agricultural land GB 15618-2018.

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
