# Peer review of "Analysis of Microbial Communities in Aged Refuse Based on 16S Sequencing"

_sustainability, doi:10.3390/su13084111_

Round 1
Reviewer 1 Report
This paper represents results of a study aimed to explore the differences in microbial composition between mineralized soil and normal soil. The selected objects are quite interesting in the context of the reclamation of aged refuse from stabilized landfills.
Nevertheless, few comments are made, which will have to be taken into account before recommendation for further publication:
Section 2.1.: it is necessary to give a more detailed description of the studied samples; the names of soils and soil horizons should be given according to the international soil classification WRB (2015).
Section 2.2. L71-74: Add specific methods that were used to determine soil fertility indicators.
L74 – soil organic carbon (SOC).
L74-77: The pH and concentrations of heavy metals cannot be determined using one method. It is necessary to separate them, indicate the names of methods and devices. Was the Hg concentration also determined using ICP-MS??!
Fig.1. A – Replace «Total nitrogen, phosphorus and potassium» with «TN, TP and TK». B – What is the difference between «organic carbon» and «organic matter? I think only in calculation. Therefore, I think, Fig. 1B needs to be removed, and the content of soil organic carbon (SOC) should be added to Fig. 1A.
Table 1. Add decoding of CSAPDA and SCALE abbreviations.
Put «mg / kg» in the table name.
The manuscript contains a lot of "more or less" reasoning, but the authors do not discuss "what is the reason?" The manuscript does not contain data on the granulometric composition, on the water regime of soils, on the position of soils in the relief... And all these factors have a very strong effect on the composition of soils! The authors provide no evidence that landfill soils and control soils can be compared at all!
Reviewer 2 Report
In this study the metagenomic 16S rRNA amplicon method was used to compare the microbial composition and diversity of aged refuse and pollution-free and unfertilized soils in order to come with new solutions for closed landfills treatment. The submitted manuscript is interesting, original and within the scope of the journal, but some changes should be addressed:
- In the introduction (lines 34-35), the authors highlighted the importance of municipal solid waste reduction which become an urgent issue. It will be interesting to update the state-of-the-art with some information regarding different methods (conventional and unconventional) for the waste treatments or reuse (please see: DOI: 10.5593/SGEM2016/B31/S12.107 and DOI: 10.5593/SGEM2016/B31/S12.007)
- I recommend to use “±” before standard deviation values presented in table 1.
- Please use in the text “figure” instead of “fig.” because you come with both variants (lines 129, 136, 138, 143…).
- Please use the same font for the axes title and scale for figure 4 A, B, C, D and try to increase their size in order to make them more readable.
- Please try to short the figures titles where is possible and come with the comments in the text.
- Please try to develop the conclusion section by adding briefly certain findings or your own remarks.
Reviewer 3 Report
The paper entitled Analysis of microbial communities in aged refuse based on 16S sequencing is focused on the differences in microbial composition between the mineralized soil and normal soil. The paper is suitable for publication in Sustainability nevertheless the following points should be improved:
1. Abstract: In last sentence, please indicate in some words for what have the association analysis of microbial communities and physicochemical indicators.
2. Material and Methods
2.1. Why authors taken only three samples of each soils type? What is the square of landfill? How close were the aged refuse soils sampling sites to each other? Please indicate more in details the soil sampling.
2.2. L76: Inductively coupled plasma mass spectrometry (as well as other analytical methods, such as chromatography, spectrophotometry etc.) should be without quotes, and the word method could be deleted.
2.3. L81: electrode can be used for measuring, but it is not the method. Please check and revise the method name.
2.4. L74-77: It is not clear, what protocols were used for heavy metal determination? Did authors analysis of heavy metals as described in [18] or in [19]?
3. Results
3.1. Figure 1: According to the International System of Units (SI) kilo should be abbreviated as “k” 9small letter), so it should be mg/kg, and nor mg/Kg. Please check and correct on figure and along manuscript.
3.2. Why on Figure 1 the aged refuse and control groups are referred as “group A” and “group B”, and on other Figures as “CK” and “T”. Please unify along all manuscript.
3.3. L 153: Please revised the phrase “the control aged refuse group” to “the control and aged refuse group”
3.4. Table 1: The data in Table 1 duplicate the data in Figure 1 C. The duplication should be avoided. Additionally, the text in lines 162-165 “As shown in Table 1, the Chao1 and ACE indices were higher in 162 the aged refuse group, while the Simpson index was lower in the aged refuse group in 163 comparison to that of the control group, indicating higher diversity of the microbial com- 164 munity in the aged refuse” does not reflect the data presented in Table 1. Please check and correct.
3.5. Subsection 3.5: Was the difference in relative abundances of the pathways between two groups significant? I am not sure that ”slightly” higher can be considered as significant (…abundances of "Metabolism of Cofactors and Vitamins" and "Energy Metabolism" were slightly higher). Additionally, I recommend the revised the phrase in Abstract reflecting these results (L19-21). In present version the phrase in abstract does not correspond to the results.
4. Discussion: I would like to invite authors to discuss more eco-physiological aspects about different strategies in alleviation from soil polluted ecosystems and about molecular mechanisms that play the crucial role in microorganism in these strategies.
5. Conclusion: Please add some aspects of practical implementation of obtained results.
Reviewer 4 Report
Dear Authors, Over the last several decades, an alarming increase in the amount of municipal solid waste has been observed, which has a significant impact on human life and the surrounding natural environment. The manuscript presented to me for review undoubtedly touches upon an innovative and very important research topic. I believe that after introducing a few changes and additions, it can be published in Sustainability. My comments: 1. The main issue is the lack of adaptation of the publication to the editorial requirements of the Sustainability (follow the template file carefully in Instructions for Authors).2. The Abstract is structured correctly, maybe a bit too long. 3. The Keywords should be in alphabetical order. 4. Introduction is very poor. The role of microorganisms in shaping the processes taking place in various types of waste should be discussed in more detail. It is necessary to list the types of these microorganisms and the mechanisms on the basis of which these processes occur. 5. Materials and Methods are briefly but correctly presented. 6. The results are presented in an interesting way, but the readability and standardization of the Figures descriptions should still be worked on. 7. In Discussion, the authors rely mainly on the latest literature, but References must undergo a thorough correction by adapting to the journal's standards. 8. The Conclusions is correct.
Round 2
Reviewer 1 Report
The authors corrected most of the comments and the manuscript has improved markedly. However, the authors did not correct 2 comments properly:
- Section 2.1.: the names of soils and soil horizons should be given according to the international soil classification WRB (2015).
- Section 2.2.: the authors have added national standards, but they are not always available to a wide range of readers. Therefore, it is necessary to briefly describe the methods (device, conditions, etc.).
Reviewer 3 Report
The authors improved the manuscript well. I am satisfied with most of the authors' responses to comments, but I have still some remarks:
1) Abstract L 61-64 and along manuscript: Please unify the letter case in phrases “Carbohydrate metabolism”, "amino acid metabolism", “Xenobiotics Biodegradation and Metabolism”, "Lipid Metabolism" etc.
2) “The association analysis of microbial communities and physicochemical indicators of the two soil groups showed that multiple environmental factors had significant impact on microorganisms abundance”. In my opinion the conclusion that “multiple environmental factors had significant impact on microorganisms abundance” is too general. In order for the conclusion to be more reflective of the results of the presented study, it is necessary to indicate specific environmental factors. I suggest the authors could specify future perspectives and practical implementation of results in Abstract and Conclusion Sections.
3) Bacteria names are always written in italics. For example, L675-676. Please check along manuscript and correct.
